# Evolution of Symmetrical Grain Boundaries under External Strain in Iron Investigated by Molecular Dynamics Method

**Wenxue Ma** [1], **Yibin Dong** [1], **Miaosen Yu** [1], **Ziqiang Wang** [1], **Yong Liu** [1], **Ning Gao** [1,2,*], **Limin Dong** [3] **and Xuelin Wang** [1,*]

1. Institute of Frontier and Interdisciplinary Science and Key Laboratory of Particle Physics and Particle Irradiation (MOE), Shandong University, Qingdao 266237, China
2. Institute of Modern Physics, Chinese Academy of Sciences, Lanzhou 730000, China
3. Department of Materials Science and Chemical Engineering, Harbin University of Science and Technology, Harbin 150080, China
* Correspondence: ning.gao@sdu.edu.cn (N.G.); xuelinwang@sdu.edu.cn (X.W.)

**Abstract:** In the present work, the evolution of atomic structures and related changes in energy state, atomic displacement and free volume of symmetrical grain boundaries (GB) under the effects of external strain in body-centered cubic (bcc) iron are investigated by the molecular dynamics (MD) method. The results indicate that without external strain, full MD relaxations at high temperatures are necessary to obtain the lower energy states of GBs, especially for GBs that have lost the symmetrical feature near GB planes following MD relaxations. Under external strain, two mechanisms are explored for the failure of these GBs, including slip system activation, dislocation nucleation and dislocation network formation induced directly by either the external strain field or by phase transformation from the initial bcc to fcc structure under the effects of external strain. Detailed analysis shows that the change in free volume is related to local structure changes in these two mechanisms, and can also lead to increases in local stress concentration. These findings provide a new explanation for the failure of GBs in BCC iron systems.

**Keywords:** grain boundary; free volume; strain effect; micro-cracking; molecular dynamics

## 1. Introduction

It is well known that body-centered cubic (BCC) Fe-based steels have been extensively used in various industrial applications [1–3]. Typically, these steels are polycrystalline materials in which grain boundaries (GBs) are formed between crystallites [4]. Thus, the grain boundaries have a significant influence on the physical, mechanical and chemical properties of Fe-based materials. The investigation of their properties, including energy, structures and mechanics, presents a popular topic in the field of materials science [5,6]. For example, GB could act as a "sink" to absorb the interstitials and vacancies [7], which can decrease the number of radiation defects but can also affect the micro-structure of GB and related mechanical properties. Furthermore, external stress or strain fields, in addition to contribution from dislocations, GB sliding, migration, dislocation nucleation and failure are all involved in plastic deformation [8,9]. Since GB failure is a crucial factor for the design and application of Fe-based materials [10], it is necessary and important to study GB failure mechanisms at the atomic scale to explore the potential reasons for these phenomena.

The relationship between the atomic structure and energy of GBs has been studied extensively in the literature [11,12]. For example, Ratanaphan et al., calculated the energies of 408 grain boundaries in Fe and Mo using embedded atom method (EAM) potentials [5]. They reported that the calculated energies vary significantly with the grain boundary plane orientation but the energies do not show any distinct trends with misorientation angle or with the density of coincident lattice sites [5]. Gao et al. studied three

typical GBs in BCC iron using molecular statics (MS) simulations, ab initio density-functional theory (DFT) calculations and the simulated high-resolution transmission electron microscopy (HRTEM) method, indicating the importance of relaxing the GBs in order to further investigate the properties of GBs through a multiscale method [13]. Du et al. also reported the energy states of some GBs in $\alpha$-and $\gamma$-Fe using density-functional theory and found that the $\sum 3$ twin boundaries exhibit low interface energies [14]. In addition to these results, the deformation behavior of GBs under different conditions is also a key factor to understand the GB properties. Spearot and colleagues investigated the influence of the structure of grain boundaries (e.g., bi-crystal GBs) on deformation behavior of the system by using MD simulations, through which the roles of dislocation nucleation and emission phenomenon under uniaxial tension were recognized for aluminum [15] and copper [16] symmetric tilt grain boundaries (STGBs). Terentyev et al. studied a set of <110> tilt grain boundaries (GB) in $\alpha$-Fe with a misorientation angle varying from 26° to 141° by applying atomistic calculations and discovered grain boundary sliding is closely related to the structure determined by the misorientation angle [4]. Singh et al. reported the investigation of structure, energy and tensile behavior of niobium (Nb) bi-crystals containing symmetric and asymmetric tilt grain boundaries via molecular dynamics simulations [6]. Cui et al., further studied the <001>, <101> and <111> twist grain boundary structures in copper through molecular dynamics simulations to obtain the dependence of tensile strength on twist angle of grain boundary [17]. They reported that for a <001> and a <101> twist grain boundary, the tensile strength increases on average with an increasing misorientation twist angle [17]. However, the misorientation bears slight influence with respect to <111> twist GB structures [17]. From these results, it can be observed that various factors affect the structure and mechanical properties of GBs, highlighting the complicated nature of this topic within the material field. In order to solve these difficulties, researchers recently applied neutron diffraction method to investigate the free volume of sub-microcrystalline Ni [18], in which the anisotropic annealing of relaxed vacancies at GBs was identified, and is considered the main reason behind induction of the anisotropic length change upon annealing [18]. However, until now there are no detailed reports regarding the free volume change in GB under an external stress or strain field, thus warranting further investigated to better understand the properties of GBs.

As stated above, in addition to energy, structure and the deformation of grain boundary, the properties of free volume (FV) in grain boundaries should also be investigated in detail especially under the effect of an external stress or strain field. Generally, free volume in materials is generally defined as the maximum volume of a sphere that can be inserted between atomic sites in the system [19]. It has been reported that the free volume in grain boundary could assist the mobility of neighboring atoms, enabling GB sliding, grain rotation and GB dislocation emission [20]. By using molecular dynamics simulations through the investigation of three face-centered cubic (fcc) and body-centered cubic (bcc) metals, Sun et al. found that the free volume shrinks much faster above a critical temperature [21]. Tschopp et al. reported the dependence of free volume on the spatial correlation functions in grain boundary [20], which provides a better understanding of dislocation dissociation and nucleation in Cu grain boundaries. Furthermore, Tucker et al. investigated the effect of free volume on the stretching process of fcc copper grain boundaries and found that the free volume influences interfacial deformation through modified atomic-scale processes [22]. Wang et al. also suggested that FV may provide a site for micro-crack nucleation in bcc symmetrical grain boundaries after irradiation [23]. However, although research has been performed investigating the effect of free volume on the properties of grain boundaries, the relationship between free volume and grain boundary mechanical properties has not been reported in detail. For example, the relationship between the FV change and the stress concentration resulting in slip system activation and the role of FV change in GB failure, warrants further investigation. The results are expected to provide a comparison and validation with neutron diffraction results induced by the external stress or strain field in the future.

Therefore, in this paper, the properties of the symmetrical grain boundaries including ∑3(112), ∑3(111), ∑5(012), ∑5(013), ∑9(221), ∑11(113) and ∑17(410) in bcc iron are investigated at the atomic scale, and in particular, the influence of free volume on stress–strain properties. New mechanisms of GB failure by considering the free volume influence were explored in this work according to the simulation results. In the following, details of the simulation methods are introduced in Section 2, and the results and discussion are provided in Section 3. Finally, the conclusions are given in Section 4.

## 2. Method

In this work, molecular dynamics (MD) simulations are performed by Large scale Atomic/Molecular Massively Parallel Simulator (LAMMPS) [24] which is an open source MD software to simulate atomic interactions for a given system. The output data are analyzed by OVITO software (Version 3.5.4, OVITO GmbH, Darmstadt, Germany) [25]. Meanwhile, as stated above, 7 different grain boundaries including ∑3(112), ∑3(111), ∑5(012), ∑5(013), ∑9(221), ∑11(113) and ∑17(410) are built in bcc Fe based on coincidence site lattice (CSL) theory [26]. The GB angle of these GBs is from around 36° to 109°. Periodic boundary condition (PBC) is applied along 3 directions. In order to avoid the interaction between two GBs induced by PBC, the distance along the normal direction of GB is set to at least 14.8 nm, as listed in Table 1. A schematic of the symmetrical GB model is shown in Figure S1, provided in the Supplementary Materials. The other information for each simulation box is also included in Table 1. Atomic structures of these 7 GBs before full relaxation are shown in Figure 1. The Fe-Fe interaction is described by Fe potential developed by Mendelev et al., [27]. This potential has been well applied for grain boundary simulations [5,13,28] and is suitable for the present purpose.

**Table 1.** Parameters of symmetrical GBs used in the present work.

| NO. | GB Plane (hkl) | Sigma ($\sum$) | Simulation Box Length(Å) | Number of Unit Cells along x, y, z and Normal Direction of GB Plane | Number of Atoms |
|---|---|---|---|---|---|
| 1. | (112) | ∑3 | 121.15 × 74.2 × 209.8 | 30 × 30 × 15, Z | 162,000 |
| 2. | (111) | ∑3 | 121.2 × 148.4 × 209.8 | 30 × 30 × 30, Y | 324,000 |
| 3. | (310) | ∑5 | 85.8 × 271.8 × 271.8 | 30 × 30 × 15, Z | 270,000 |
| 4. | (210) | ∑5 | 85.8 × 191.5 × 191.5 | 30 × 30 × 15, Z | 540,000 |
| 5. | (221) | ∑9 | 121.2 × 363.4 × 256.9 | 30 × 30 × 15, Z | 972,000 |
| 6. | (113) | ∑11 | 133.9 × 80.8 × 189.4 | 10×20×20, Z | 176,000 |
| 7. | (410) | ∑17 | 235.4 × 57.2 × 235.4 | 20 × 20 × 10, Z | 272,000 |

Based on the above simulation models, conjugate-gradient (CG) method is used for relaxation at 0 K, and subsequently followed by MD relaxation at 300 K and 600 K. After full MD relaxation, the CG method is applied again to obtain the atomic structure and energy at 0 K. In this way, the local structure may overcome the energy barrier and reach a lower energy state after the CG-MD-CG relaxation process. In CG relaxation, the specified energy tolerance is $10^{-10}$ and the specified force tolerance is $10^{-10}$ eV/Å$^3$. During the MD relaxation, the timestep is 1 fs and at least 20 ps relaxation is applied for each process to ensure the system is fully relaxed at the given temperature. It should also be noted that during relaxation, the constant number of atoms, pressure and temperature (NPT) ensemble is applied in order to relax both the atomic position and simulation volume.

Furthermore, for volume relaxation, each direction of the box is allowed to relax independently to fully release the internal elastic stress and obtain a more stable state for further calculations. The GB formation energy, $E_{\mathrm{GB}}$, is then calculated according to the following equation as applied in previous studies [13,29,30], which is defined as the difference between the potential energy $E_{total}$ of $n$ atoms in the supercell containing GBs and the potential energy of a computational cell with the same number of atoms in a perfect crystal, divided by the cross-sectional area, $S$, of two GB planes.

$$E_{\mathrm{GB}} = \frac{E_{total} - NE_p}{2S} \tag{1}$$

where $E_{total}$ is the total energy of a system containing 2 GBs. $N$ is the number of atoms in this system. $E_p$ is the cohesive energy of one atom in a perfect bcc Fe crystal, which is −4.12 eV according to Mendelev potential [27].

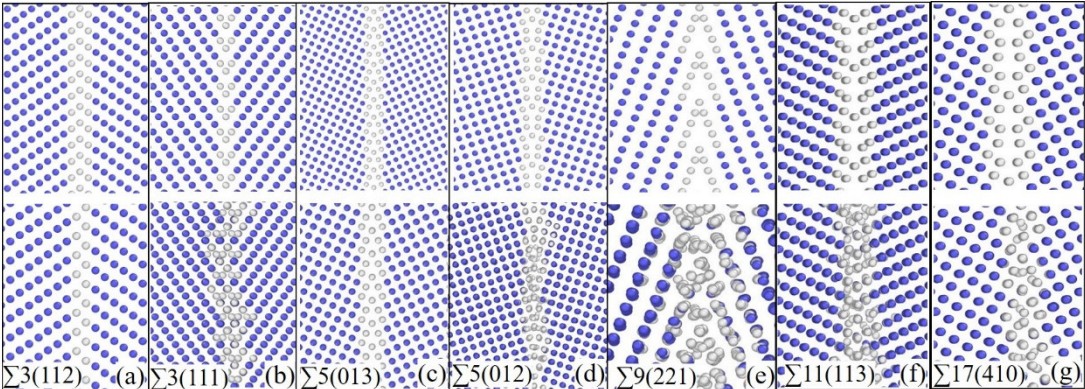

**Figure 1.** GB atomic structures before and after MD relaxations shown in the top and bottom panels, respectively: (**a**) ∑3(112) GB, (**b**) ∑3(111), (**c**) ∑5(013), (**d**) ∑5(012), (**e**) ∑9(221), (**f**) ∑11(113) and (**g**) ∑17(410). Structure types are analyzed by the common neighbor analysis method. Atoms near the GB region are colored white, while the atoms in the grain are colored blue.

To investigate the mechanical property of the grain boundary, after full relaxation, external strain is applied to the system along the normal direction of GB plane with a strain rate of $5 \times 10^9$/s. The simulation temperature is set at 300 K. It should be noted that during the strain application along the normal direction of GB plane, the box length along the other two directions of the computational box is allowed to change automatically to ensure zero pressure along these two directions. The stress related to the increases in strain is also calculated and thus, the stress–strain curve is obtained. For comparison, this process is also performed for a perfect structure with strain direction along the same normal direction of GB plane. During the strain application along GB normal direction, the length of the box along the directions perpendicular to GB normal direction is also allowed to vary, which is similar to the real experimental condition and ensures the system is at a single stress–strain state [31].

To investigate the underlying factors affecting the mechanical property of GB, different methods are used to analyze GB structures and properties including the common neighbor analysis (CNA) [32], free volume (FV), single-atom potential energy distribution, local stress field and the single-atom displacement magnitude. In this work, CNA analysis, atomic energy distribution and displacement are performed by OVITO software and details can be found in [25]. The free volume is obtained by introducing enough grids along three directions in computation box and calculating the maximum distance ($D_{max}$) of each grid point to surrounding atoms. The free volume is then calculated as the sphere volume with radius of $D_{max}$-$r$, where $r$ is the atomic radius in perfect lattice. This method has been well used in previous studies [33]. The FV calculation under the application of

external stress was performed for polymer and amorphous materials [34], which have an FV distribution through the whole system. In this work, this method is also applied firstly for GB investigation since the FV in GB is expected to change in the GB failure process. Furthermore, as stated in the Introduction, the results from the present work may be used for comparison with neutron diffraction measurements under the effect of external stress field in future. The stress of each atom is calculated according to the following equation and can be viewed via the Ovito software:

$$\sigma_{ij}^V = \frac{1}{V} \sum_{\alpha} \left[ \frac{1}{2} \sum_{\beta=1}^{N} \left( R_i^\beta - R_i^\alpha \right) F_j^{\alpha\beta} - m^\alpha v_i^\alpha v_j^\beta \right] \tag{2}$$

where ($i$,$j$) take values of x, y and z (directions). $\beta$ takes values 1 to $N$ neighbors of atom $\alpha$. $R_i^\alpha$ is the position of atom $\alpha$ along direction $i$. $F_j^{\alpha\beta}$ is the force (along direction $j$) applied on atom $\beta$ from atom $\alpha$. V is the total volume of the system. $m^\alpha$ is the mass of atom $\alpha$ and $v_i^\alpha$ is the thermal excitation velocity of atom $\alpha$ along $i$ direction.

## 3. Results and Discussion

### 3.1. Grain Boundary Energy and Structure after Different Relaxation Processes

Grain boundary energies of seven GBs calculated by using the method mentioned above are listed in Table 2, which includes the results following CG relaxation and CG-MD-CG relaxation. For comparison, the results from DFT calculations [35,36] are also listed. From this table, it is clear that CG-MD-CG relaxation results in lower energy state of GB according to Equation (1). The difference with and without MD relaxation is smaller for $\sum 3(111)$, $\sum 3(112)$ and $\sum 5(013)$ GBs while it is larger for the other cases investigated in this work. The reason for this relates to the local atomic structure change induced by MD relaxation, as indicated by CNA and displacement results, shown in Figure 2. For example, $\sum 3(112)$ and $\sum 5(013)$ grain boundaries almost retain the symmetrical character with short atomic displacement distances after MD relaxation. However, for $\sum 3(111)$ GB, after 600 K MD relaxation, some atoms located in the grain boundary area have moved a long distance, up to 1.3 Å, resulting in the loss of symmetry locally. In contrast, $\sum 5(012)$, $\sum 9(221)$, $\sum 11(113)$ and $\sum 17(410)$ GBs have almost lost the symmetrical character at the GB plane region as shown by displacement of the atoms in the GB region. The maximum atomic displacement of these cases is up to 4.81 Å. Therefore, the high temperature MD relaxations induced losses in symmetrical features in certain symmetrical GBs, through which these systems have overcome the energy barriers to lower energy states. In fact, a similar conclusion was made in previous studies [23,33], while in this work, more GBs confirm that the MD relaxation to a lower state is necessary for further analysis. The relationship between the GB formation energy and GB angle has also been investigated, as shown in Figure 2, which can be fitted by a Gaussian function (Equation (3)). Further, we have defined a new variable called $\Delta\sigma$, which is a fit for stress peak value by GB energy, as shown in Equation (4). $\Delta\sigma$ could help explain how the free volume had coupled with stress in the GB failure process.

$$E_{\text{GB}} = 1.228 - 0.958\text{exp}\left[ -0.5\left( \frac{\theta - 109.49}{5.1696} \right)^2 \right] \tag{3}$$

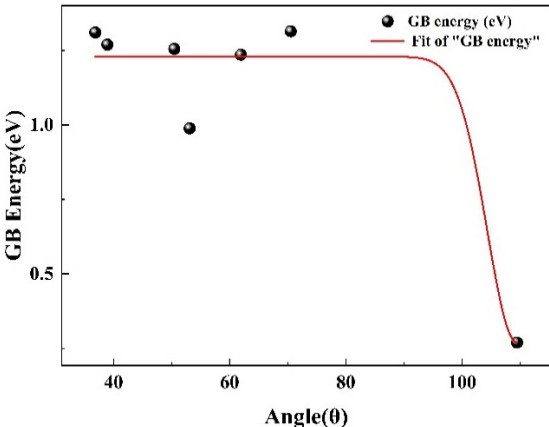

**Figure 2.** Gaussian Fit of the relationship between GB energy and GB misorientation.

**Table 2.** Grain boundary energy calculated by CG and CG-MD-CG relaxation processes. For comparison, the results from DFT calculations are also listed.

| Sigma (∑) | $E_{GBs}$ J/m² (CG-MD-CG) | $E_{GBs}$ J/m² (CG) | DFT |
|---|---|---|---|
| ∑3(112) | 0.2703 | 0.3233 | 0.46 [35] |
| ∑3(111) | 1.3144 | 1.3438 | 1.61 [36] |
| ∑5(012) | 1.1301 | 1.6741 | 1.64 [35] |
| ∑5(013) | 0.9889 | 1.0649 | 1.6 [35] |
| ∑9(221) | 1.2639 | 2.3318 | 1.66 [37] |
| ∑11(113) | 1.2551 | 2.4361 | 1.45 [37] |
| ∑17(441) | 1.2356 | 2.5315 | - |

### 3.2. Evolution and Related Failure Mechanisms of GBs under the External Strain Effect

In this section, the stress–strain curves for GBs and related single perfect crystal cases are calculated firstly, as shown in Figure 3. It is clear from Figure 3a, ∑3(111) and ∑17(410) GBs have the maximum and minimum peak tensile stress, respectively. In fact, a similar conclusion could also be made for single perfect crystal cases, as shown in Figure 3b. The peak tensile stresses of all cases studied in this work are listed in Table S1, in the Supplementary Materials. Compared with the results of single crystals, it is clear that the appearance of GBs induces the decrease in peak tensile stress.

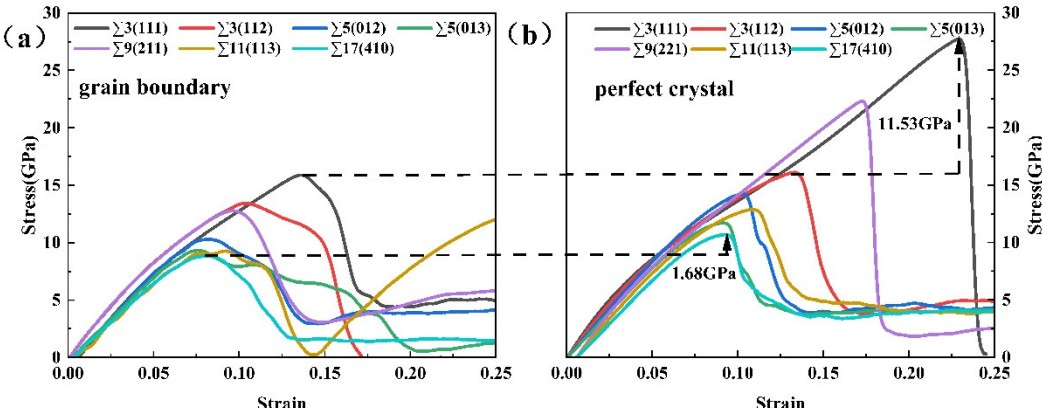

**Figure 3.** (**a**) Strain–stress curves of ∑3(111), ∑3(112), ∑5(012), ∑5(013), ∑9(221), ∑ 11(113), ∑17(410). (**b**) Strain–stress curves of single crystals corresponding to each grain boundary.

In order to explore the reason behind the peak value decrease induced by GB defect, the atomic stress state and related atomic structure were analyzed for these GB systems. Based on the analyses, the following mechanisms have been explored.

### 3.2.1. Phase Transformation Induced Grain Boundary Failure

Figure 4 depicts the structural evolution of the $\sum3(112)$ and $\sum5(013)$ GB system under the effect of external strain. From these results, it can be found that with increasing strain, the phase transition occurs initially in the GB region and extends to the grain interior, that is, from an initial bcc phase to fcc phase, as shown in Figure 4a. In the present work, this transition occurs with strain around 10% and related stress around 12.67 GPa, as shown in Figure 3a, and is also confirmed by the atomic potential energy and displacement, that is, atoms in the phase-transition region have higher potential energy and larger displacement distance, as shown in Figure S2 in the Supplementary Materials. In fact, the phase transition from bcc to fcc under external stress has also been reported in polycrystalline Fe systems with stress values up to around 13 GPa under shock wave [38]. It is clear that $\sum3(112)$ GB has a similar effect on the phase transformation of Fe system. An example of a formed fcc structure is also shown in Figure S3 in the Supplementary Materials with a lattice constant around 3.7 Å, which is slightly larger than the value (3.6 Å) of fcc-Fe under the normal condition [39]. Together with this phase transition, a new interface is formed between the newly formed fcc phase and the original bcc phase, as shown in Figure 4b, whose position varies with the phase transition until the slip system {123}<111> becomes activated near the interface from the bcc phase side. Dislocation nucleation is then initiated from this activated slip system and finally the dislocation network is formed with increasing strain. One possible reason for the slip system activation in bcc instead of fcc phase for $\sum3(112)$ GB can be explained by different Schmid factors ($\mu$) in these two phases with external stress along <112> direction. In this case, the maximum Schmid Factor $\mu$ of bcc and fcc slip systems is 0.4115 and 0.4082, respectively. The larger Schmid factor in bcc phase indicates the it has higher probability to initiate the slip system in bcc phase. The value of 0.4115 in bcc phase is related to the {123}<111> slip system, same as the present simulation results, as shown in Figure 4b. In addition to above results, the larger displacement related to the phase transition is also confirmed to result in a larger free volume near the bcc-fcc interface, which is suspected as a possible reason for the change in stress field of the GB system. For example, comparing the maximum free volume in GB at states of 0 strain and 15% strain, as shown in Figure 5, it is clear that without strain, the maximum free volume ($FV_{max}$) is around 17.99 Å$^3$ and the average value of free volume is around 7.99 Å$^3$. While with 15% strain, $FV_{max}$ reaches around 25.71 Å$^3$ with an average value around 11.98 Å$^3$ at the bcc-fcc interface induced by the phase transition. The stress field around this maximum free volume is around 21.48 GPa, resulting in the local stress concentration increasing and related activation of the slip system. Thus, the present results indicate that the increase in free volume induced by phase transition may be also one possible reason to induce the local stress to its critical value and the failure of GB system in bcc Fe.

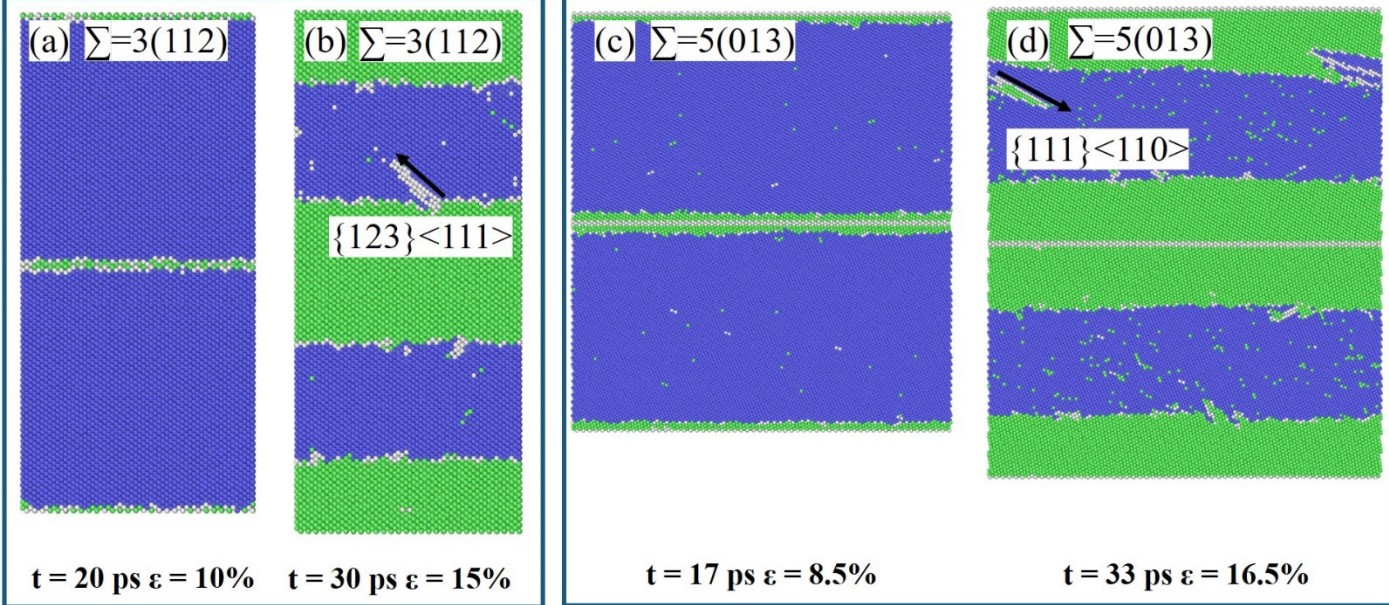

**Figure 4.** Snapshots of ∑3(112) and ∑5(013) GB under external strain effect. (**a**) Shows the state of ∑3(112) GB at which the phase transition starts with a strain value of 10% and (**b**) is the state at which the slip system in bcc phase near bcc-fcc interface is activated for ∑3(112). (**c**) Shows the state of ∑5(013) GB at which the phase transition starts with a strain value of 8.5% and (**d**) is the state at which the slip system is activated for ∑5(013) in fcc phase near the bcc-fcc interface. In the figure, the green, blue and white points are atoms in fcc, bcc and other phase states, respectively.

In fact, a similar process has also been observed in the ∑5(013) case, as shown in Figures S4 and S5 in the Supplementary Materials. The difference between these two cases is that in ∑5(013) case, there are still three atomic layers at the GB center without going through the phase transition process, as shown in Figure 4c. The phase transition occurs with strain up to 8%, at which the maximum free volume is also observed near the bcc-fcc interface with $FV_{max}$ up to 24.43 Å$^3$. The local stress concentration is also observed above the maximum free volume region with a value around 21.65 GPa, resulting in the activation of slip system and related dislocation nucleation, as shown Figure 4d. Different to the activation of slip system initially in bcc phase for ∑3(112) GB, the activation of slip system under external stress along the <013> direction occurs initially in fcc phase in the {111} plane along <110> direction. Following the same method, the Schmid factor is also calculated for this case with external stress along <013> direction. The maximum Schmid factor for bcc and fcc phases are 0.4115 and 0.4899, respectively, in this case, which is the main reason for slip system activation near the interface from fcc phase side, as shown in Figure 4d.

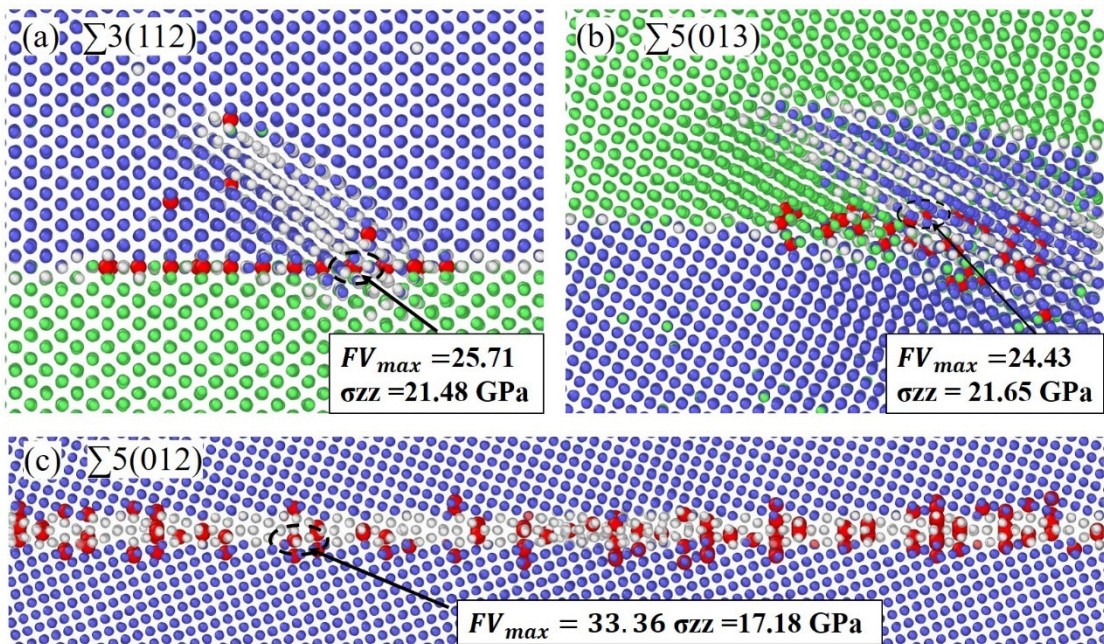

**Figure 5.** The maximum free volume and related stress distribution near the bcc-fcc interface when the stress is up to the peak value (shown by stress-strain curve in Figure 2 for (**a**) $\sum 3(112)$, (**b**) $\sum 5(013)$ and (**c**) $\sum 5(012)$ GB respectively. The blue and green balls are atoms in bcc and fcc state respectively. The larger red balls are free volume higher than 20 Å³ in GB region.

3.2.2. Mechanical Failure Induced by Activation of Slip System from GB Plane

The second phenomenon accompanying the failure of GB explored in this work is the activation of slip systems directly at GB region without going through the phase transformation, as observed in the $\sum 5(012)$ and $\sum 3(111)$ $\sum 9(221)$, $\sum 11(113)$ and $\sum 17(410)$ GB cases. One example of $\sum 5(012)$ case is shown in Figure 6. As shown in Figure 6, the slip systems are activated from the GB plane to the grain interior at a time of around 16 ps after applying the external strain, at which the strain is around 8%. The Schmid factor is also calculated for this case. The results indicated the $\mu_{max}$ is up to 0.4625, which is related to the slip in {123} plane along <111> direction, as shown in Figure 6c. Furthermore, careful analysis of the local structure indicates local disordered regions in the GB region, which have high potential energy and high stress along the normal direction of GB plane. In fact, the local stress concentration is also observed in these regions with maximum stress around 27.4 GPa, as shown Figure S6 in the Supplementary Materials. Following the analysis method in Section 2, the atomic displacements were calculated around GB, indicating the maximum displacement distance also observed in these regions, as shown in the Supplementary Materials. The free volume change around these regions is then calculated. When the strain is 0, FV$_{max}$ is around 31.24 Å³, which is then increased to 33.36 Å³ above the disordered region, as shown in Figure 5c Therefore, the maximum free volume change is also one possible factor relating to the failure of $\sum 5(012)$ GB from the activation of slip system directly in local GB plane region. Further analysis of $\sum 3(111)$ $\sum 9(221)$, $\sum 11(113)$ and $\sum 17(410)$ GB reaches a similar conclusion. The example of $\sum 3(111)$ GB has been shown in Figures S7 and S8 in the Supplementary Materials. Based on these results, the derivative of critical stress along the normal direction of GB plane of GB failure, $\Delta\sigma$, can be described as a function of GB formation energy, $E_{GB}$, as shown in Figure 7a and the following equation:

$$\Delta\sigma = 1.241 - 1.297 exp\left[-0.5\left(\frac{E_{GB} - 4.073}{0.3886}\right)^2\right] \tag{4}$$

Furthermore, the dependence of critical stress ($\sigma + \Delta\sigma$) of GB failure on free volume change ($\Delta FV$) has also been explored, as shown in Figure 7b, which can be described by a

new equation (Equation (5)). All of these results indicate that once the free volume change has been identified, the mechanical properties of GB can be estimated.

$$\sigma + \Delta\sigma = 8.21\,\Delta FV^{0.2421} \tag{5}$$

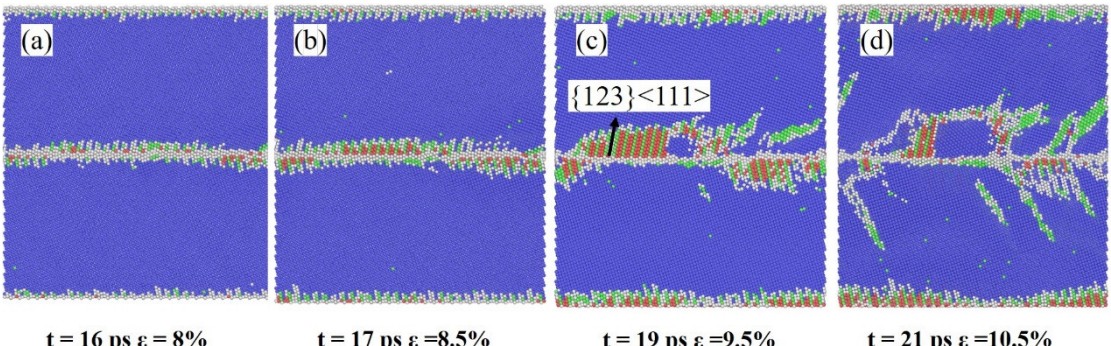

t = 16 ps ε = 8%  t = 17 ps ε = 8.5%  t = 19 ps ε = 9.5%  t = 21 ps ε = 10.5%

**Figure 6.** Snapshots of ∑5(012) GB evolution under external stress at different simulation times (t): (**a**) t = 16 ps, (**b**) t = 17 ps, (**c**) t = 19 ps and (**d**) t = 21 ps, respectively.

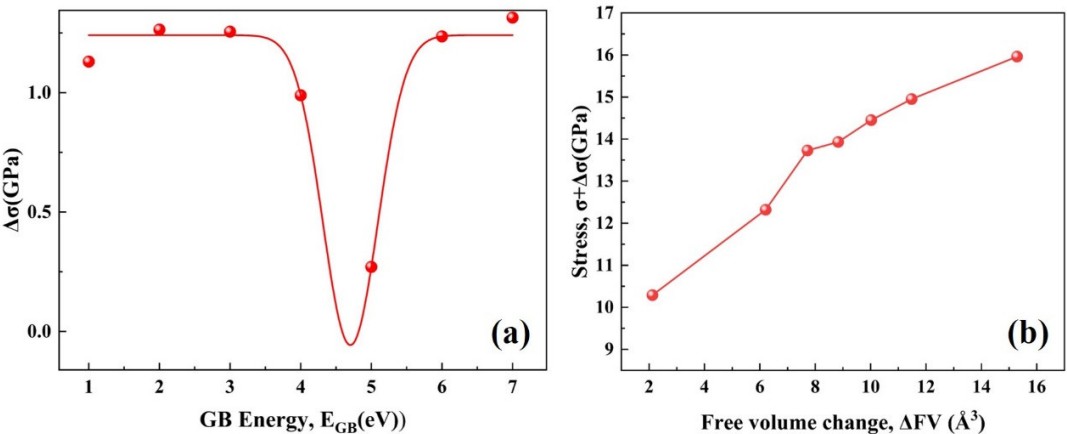

**Figure 7.** The Gaussian Fit of the relationship between GB energy and $\Delta\sigma$ (**a**). The relationship between free volume and fit stress. Fit stress equals the sum of the stress peak value and $\Delta\sigma$ (**b**).

## 4. Conclusions

In this work, the failure mechanism of symmetrical grain boundaries (GB) under external strain effects in body-centered cubic (bcc) iron are investigated via the molecular dynamics (MD) method. The local atomic structure evolution, energy state change, atomic displacements and free volumes were calculated for the above purpose. The following conclusions have been made:

(1) Full MD relaxations at high temperatures are necessary to obtain the lower energy states of GBs for further simulations under external strain.

(2) Two mechanisms are explored for the failure of symmetrical GBs under the external strain effect, including slip system activation, dislocation nucleation and dislocation network formation initially from the GB plane region induced by the external strain field or from the bcc-fcc phase interface induced by phase transformation under external strain effects.

(3) The change in free volume near the GB plane or bcc-fcc interface is not only related to the local structure change in the above two mechanisms, but can also lead to increases in the local stress concentration, providing a new explanation for the failure of GBs in BCC iron system.

**Supplementary Materials:** The following supporting information can be downloaded at: www.mdpi.com/article/10.3390/met12091448/s1, Figure S1: Schematic of grain boundary simulation

model used in the present work. For stress-strain simulations, the external tensile strain field is applied on top and bottom surface of box, Figure S2: Atomic potential energy (**a**) and displacement distribution (**b**) of ∑3(112) GB at state with peak stress, Figure S3: Example of FCC lattice structure in fcc phase after phase transition, Figure S4: Snapshots (y-z plane) shows ∑ = 5(013) undergoes phase transition and green atom is fcc struc-ture blue atom is bcc. Then the GB happens to crack at 31 ps, region a, b and c are most obvious, Figure S5: Atomic potential energy (**a**) and displacement distribution (**b**) of ∑5(013) GB at state with peak stress, Figure S6: (**a**) The structure of ∑5(012) when slip system is activated with strain around 8%. The potential energy(b), stress(c) and atomic displacement distribution(d) at this state are shown respectively. Figure S7: (**a**) The structure of ∑3(111) when slip system is activated with strain around 14%. The potential energy, stress and atomic displacement distribution at this state are shown in (**b**), (**c**) and (**d**) respectively, Figure S8: Distribution of free volume near ∑3(111) GB region at (**a**) 0 ps and (**b**) at 28 ps (strain around 14%), Table S1: The peak tensile stresses of all cases studied in this work are listed in table.

**Author Contributions:** Conceptualization, N.G.; methodology, W.M. and N.G.; software, W.M.; validation, W.M.; formal analysis, W.M.; investigation, W.M.; resources, N.G.; data curation, W.M.; writing—original draft preparation, W.M.; writing—review and editing, Y.D., M.Y., Z.W., Y.L., N.G., L.D. and X.W.; visualization, W.M.; supervision, N.G.; project administration, N.G.; funding acquisition, N.G., X.W. and Y.L. All authors have read and agreed to the published version of the manuscript.

**Funding:** This research was funded by the National Natural Science Foundation of China, grant number (Project Nos. 12075141, 12175125 and 12105159).

**Institutional Review Board Statement:** Not applicable.

**Informed Consent Statement:** Not applicable.

**Data Availability Statement:** The data presented in this study are contained within the article.

**Conflicts of Interest:** The authors declare no conflict of interest.

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
