# Peer review of "Evolution of Symmetrical Grain Boundaries under External Strain in Iron Investigated by Molecular Dynamics Method"

_metals, doi:10.3390/met12091448_

Round 1
Reviewer 1 Report
The authors present molecular simulations study of grain boundaries in bcc iron modelled by the embedded atom method. They investigate the effect of applying strain to the grain boundary. While the general approach appears valid and results presented relatively well, there are several issues that need to be addressed before I can recommend this paper for publication.
1. For CG-MD-CG process, the MD was done in NPT. Was this an isotropic NPT (i.e. only volume was changing, while the ratios of simulation box lengths remained constant)? Or was it anisotropic, so that each direction would be allowed to relax independently. Given that the system is anisotropic, the latter would be preferable. Please clarify.
2. The NPT simulation will lead to a change in the simulation box size. In Table 1, only one system size is given, which I assume is for the first CG stage. How does the size change after the MD stage? Also, during the second CG stage, what happens to the simulation box? In fact, what is the density of the bcc crystal that you are simulating? Since the crystal expands with increasing temperature, does the density correspond stress-free crystal at 0K, 300K, or 600K (in the CG stages?)
3. When you apply the strain in direction normal to the grain boundary, you allow the tangential directions to change size. How do you do it? Do you use the anisotropic NPT simulation and keep zero pressure in the tangential directions? Please give details.
4. In the strained state, part of the bcc crystal transforms to fcc. Can you please report the densities of the bcc and fcc phases in this case?
5. In Figure 1, you only explain the colors of blue and white atoms, but I also see a red atom in (d). What is it? Also in Figure 4, you say that the colors are explained in the caption to Figure 3, but it does not explain the meaning of yellow, orange, red colors. Please explain.
6. Some language issues need to be fixed. For example, when you say in the title and throughout the paper 'under external strain effect', it is better to just say 'under external strain'.
Author Response
Response to Reviewer
We thank the referee sincerely for your careful reading and valuable comments and questions. We have answered all the questions point-by-point and corrected the paper accordingly. We provide our detailed responses (in blue font) to the comments of the referees (in black font) in the following pages. The red highlighted text in the revised version indicates where the changes have been made as required by Editor. We hope these answers and corrections would satisfy the referee.
Response to Reviewer 1
Comments and Suggestions for Authors:
The authors present molecular simulations study of grain boundaries in bcc iron modelled by the embedded atom method. They investigate the effect of applying strain to the grain boundary. While the general approach appears valid and results presented relatively well, there are several issues that need to be addressed before I can recommend this paper for publication.
Answer: We sincerely appreciate referee reviewing the manuscript! We are also very pleased that you give us positive comments on our research. We have answered all your questions, and the response to each comment is presented point-to-point in the following sections.
Comments 1: For CG-MD-CG process, the MD was done in NPT. Was this an isotropic NPT (i.e. only volume was changing, while the ratios of simulation box lengths remained constant)? Or was it anisotropic, so that each direction would be allowed to relax independently. Given that the system is anisotropic, the latter would be preferable. Please clarify.
Answer: Thank referee for this question. As you expected, the anisotropic relaxation is performed for CG-MD-CG process in this work, which allows the independent change of box length along normal and parallel directions of GB plane. Following your suggestion, we have included the above information in the revised version on page 3.
On page 3
“Furthermore, for volume relaxation, each direction of box is allowed to relax independently to fully release the internal elastic stress and obtain the more stable state for further calculations.”
Comments 2: The NPT simulation will lead to a change in the simulation box size. In Table 1, only one system size is given, which I assume is for the first CG stage. How does the size change after the MD stage? Also, during the second CG stage, what happens to the simulation box? In fact, what is the density of the bcc crystal that you are simulating? Since the crystal expands with increasing temperature, does the density correspond stress-free crystal at 0K, 300K, or 600K (in the CG stages?)
Answer: Thank referee for pointing this out. It is true that the information given in the original manuscript is the initial simulation box we build. Following your suggestions, we included the box size information during the CG-MD-CG process, as shown in the following Table A1, in which the average length along each direction is listed.
Furthermore, the density of the stress-free bcc crystal is 7.96 g/cm3 at 0 K, 7.92 g/cm3 at 300 K and 7.86 g/cm3 at 600 K in the CG-MD-CG process. After including the GBs in box, the density of bcc phase is 7.96 g/cm3 at 0 K 7.93 g/cm3 at 300 K and 7.86 g/cm3 at 600 K, respectively, which is very close to the value at stress-free state.
Table A1 The box size information during the CG-MD-CG process.
|
NO. |
GB |
Simulation box length(Å)CG1 |
Simulation box length(Å)MD |
Simulation box length(Å)CG2 |
|
1. |
∑3(112) |
121.15×74.2×209.8 |
121.6×74.5×210.7 |
121.13×74.2×209.8 |
|
2. |
∑3(111) |
121.2×148.4×209.8 |
121.78×149.15 ×210.9 |
121.29×148.55×210.09 |
|
3. |
∑5(013) |
85.7×271.02×271.02 |
86.0×272.19×272.19 |
85.7×271.02×271.02 |
|
4. |
∑5(012) |
85.7×191.5×191.5 |
86.1×192.5×192.5 |
85.7×191.69×191.69 |
|
5. |
∑9(221) |
121.24×363.72×257.19 |
121.72×365.16×257.2 |
121.17×363.56×257.05 |
|
6. |
∑11(113) |
134.18×80.9×189.7 |
134.61×81.17×190.37 |
134.01×80.81189.52 |
|
7. |
∑17(410) |
235.4×57.2×235.4 |
236.6×57.4×236.6 |
235.59×57.13×235.59 |
Fig.A1 The change of box size during MD relaxation process from 0K, 300 K up to 600 K and finally to 0 K process. The relaxation time for each relaxation process is 20 ps.
Comments 3: When you apply the strain in direction normal to the grain boundary, you allow the tangential directions to change size. How do you do it? Do you use the anisotropic NPT simulation and keep zero pressure in the tangential directions? Please give details.
Answer: Thank referee for these questions. Yes, we took the way as you expected. In this work, we used Lammps to perform the simulation of box relaxation under NPT. In this software, one method has been included by applying external strain along a given direction but keeping the zero pressure along its two perpendicular directions. In this way, the system is in a state with stress only along given direction and the other two directions are stress-free by changing the box length automatically according to the potential energy surface. Following your suggestion, we provided the details of this method in the revised version on page 4.
On page 4
“It should be noted that during the strain application along the normal direction of GB plane, the box length along the other two directions of the computational box is allowed to change automatically to ensure the zero pressure along these two directions.”
Comments 4: In the strained state, part of the bcc crystal transforms to fcc. Can you please report the densities of the bcc and fcc phases in this case?
Answer: Thank referee for this question. Following your suggestion, we provide the information of density of bcc and fcc phase along the phase transition process. Furthermore, the ratio of bcc to fcc phase is also provided in the revised version. From these results, it is clear that phase transition starts at 19 ps, and the concentration of fcc phase is up to 56.4% at the 31 ps, then it decreases rapidly with the slip system activation.
Fig.A2 (a) Phase concentration with time under the external strain for ∑3(112) identified by CNA method; (b) Density of bcc and fcc phase in phase transition process.
Comments 5: In Figure 1, you only explain the colors of blue and white atoms, but I also see a red atom in (d). What is it? Also in Figure 4, you say that the colors are explained in the caption to Figure 3, but it does not explain the meaning of yellow, orange, red colors. Please explain.
Answer: Thank referee for pointing this out. We apologize for missing the related information. In this work, we used Ovito for visualization of 3D atomic structure of system. When we checked randomly the state of atom in GB region, we used the selection function of Ovito, which would show the red color after the selection. After checking its property, we forgot to release it and left it as a red atom. In the revised version, we have updated the Fig.1(d) by showing the appropriate color of all atoms. The explanation of atomic color in Fig.6 (the Fig.4 in the original version) has also been included in revised version on page 5.
On page 5
Figure 2. GB atomic structures before and after MD relaxations shown in the top and bottom panels respectively: (a) ∑3(112) GB, (b)∑3(111), (c)∑5(013), (d)∑5(012), (e) ∑9(221), (f)∑11(113) and (g) ∑17(410). Structure types are analyzed by common neighbor analysis method. The atoms near the GB region are colored white, while the atoms in the grain are colored blue.
On page 8
Figure 5. The maximum free volume and related stress distribution near the bcc-fcc interface when the stress is up to the peak value (shown by stress-strain curve in Fig.2 for (a) ∑3(112), (b) ∑5(013) and (c) ∑5(012) GB respectively. The blue and green balls are atoms in bcc and fcc state respectively. The larger red balls are free volume higher than 20 Å3 in GB region.
Comments 6: Some language issues need to be fixed. For example, when you say in the title and throughout the paper 'under external strain effect', it is better to just say 'under external strain'.
Answer: Thank referee for pointing this out. Following your suggestion, we read manuscript carefully and correct the possible language issues in the revised version. For example, the question you asked about the title has been revised.
On page 1
“Evolution of symmetrical grain boundaries under external strain in iron investigated by molecular dynamics method”

Reviewer 2 Report
Evolution of symmetrical grain boundaries under external strain effect in iron investigated by molecular dynamics method
In this study, the author claims that relaxation at 0K needs to be performed after running a MD simulation for getting the minimum energy structure. According to my view this claim of author is quite intuitive as the atoms would have moved under the influence of temperature and needs to be relaxed for getting the lowest energy state. Moreover, the claim the author is making about the difference in the energies of the GBs before and after MD simulation is also obvious. One of the reasons is stated by the author that is distortion and another reason according to my thought process is the change in the lattice parameter after running at high temperature. Change in the lattice parameter would affect the energy of the system. Thus, I don’t see any new science coming out from the first part of the results.
Author should provide more detail about the various parameters used for relaxing and running the MD simulation, such as energy and force criteria, time step, etc.
The author does not do a great job of setting up the stage for the problem. The introduction needs to be re-written for setting up the stage.
Overall, author should compare the methodology proposed is new and valuable by comparing with the previous literature work.
Could author provide more detail about the Schmid factor and how is it obtained?
Furthermore, there are minor error such as inconsistency in using et al., figures are not in place, etc. Please review these changes as well.
Reviewer 3 Report
The manuscript entitled "Evolution of symmetrical grain boundaries under external strain effect in iron investigated by molecular dynamics method" studies the evolution of atomic structure and related changes of energy state, atomic displacement, and free volume of symmetrical grain boundaries under external strain effect in bcc iron. The work is well-written and well-organized. However, it should be rejected or considerably revised. The subject is not really new, there is plenty of work on GBs in iron with different lattices, different orientations, etc. The authors did nice and honest work, and studied all the processes in detail, but I do not feel that it is interesting for most of the readers of this journal. The subject is very special and the results are not compared properly with that from the literature. The Introduction is weak since it can be seen that this topic was popular years and years ago. It is not clear from the Introduction why they need to study such GBs in iron. The Authors mentioned several works, but just describe them, not doing some conclusions. It is seen that a good job is done, it is rigorous and accurate, but this publication is more suitable for a journal where MD simulations are presented.
Round 2
Reviewer 1 Report
The authors addressed my comments and concerns, so the paper can now be published.
Reviewer 2 Report
The authors did a good job of working on the comments.